# Feudal Reinforcement Learning by Reading Manuals

## Abstract

Reading to act is a prevalent but challenging task which requires the ability to reason from a concise instruction. However, previous works face the semantic mismatch between the low-level actions and the high-level language descriptions and require the human-designed curriculum to work properly. In this paper, we present a Feudal Reinforcement Learning (FRL) model consisting of a manager agent and a worker agent. The manager agent is a multi-hop plan generator dealing with high-level abstract information and generating a series of sub-goals in a backward manner. The worker agent deals with the low-level perceptions and actions to achieve the sub-goals one by one. In comparison, our FRL model effectively alleviate the mismatching between text-level inference and low-level perceptions and actions; and is general to various forms of environments, instructions and manuals; and our multi-hop plan generator can significantly boost for challenging tasks where multi-step reasoning form the texts is critical to resolve the instructed goals. We showcase our approach achieves competitive performance on two challenging tasks, Read to Fight Monsters (RTFM) and Messenger, without human-designed curriculum learning.

## 1 Introduction

Recently, there are increasing interests in building reinforcement learning (RL) agents that interact with humans via natural language, such as follow natural language instructions and complete goals specified in natural language. The successes of these studies will boost the user experience in a wide range of real-world applications, such as visual language navigation (Anderson et al., 2018; Wang et al., 2019b), interactive games (Gray et al., 2019), robot control (Tellex et al., 2020), goal-oriented dialog systems and other personal assistant applications (Dhingra et al., 2017).

In order to generalize to real-world use cases, the research of RL with language instructions faces various kinds of complexity. One critical demand of these use cases is that humans tend to give concise instructions, which specify the goals they hope to achieve, instead of providing complete information for the intermediate steps. For example, users would prefer to ask a personal assistant "*share my recent photo to my parents*". But if the agent only works when specified the unusually complex instruction "*open the photo albums, select the recent one and click on sharing, then click Messages and select the contacts named mom and dad*", it will significantly lower the users' satisfaction.

To deal with this challenge, it is important for an agent to reason a procedure to accomplish the goal, with the latent unspecified information supplemented with necessary prior knowledge about the environment. One natural and realistic source of such knowledge is the textual descriptions about the environment and its dynamics, e.g., the textual manual of the environment. This setting gives rise to a new problem of **reading to act**, where the agents require to making actions via comprehension of both the manuals and the environments, then bridging them together. Recently, this direction draws increasing attention with new benchmarks established (Branavan et al., 2012; Narasimhan et al., 2017; Zhong et al., 2020; Wang & Narasimhan, 2021).

While progresses have been made in the field of reading to act, there is one fundamental challenge not studied much, i.e., the **semantic mismatch** between the *low-level actions and states agents have access to* and the *high-level language descriptions in the instructions and manuals*. Specifically, in the example shown in Figure 1, it is easy for human players to derive a sequence of steps to

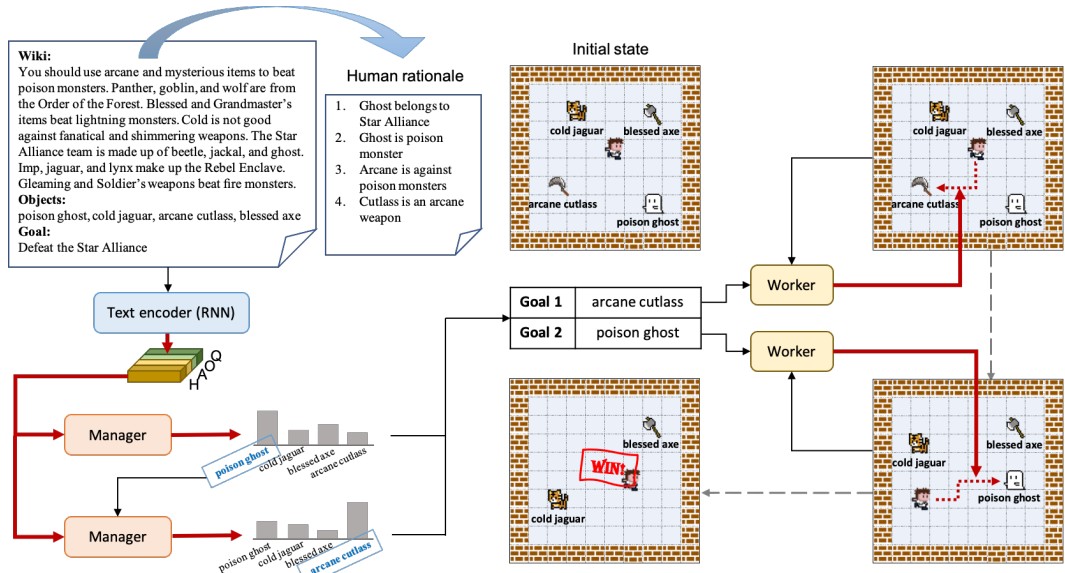

Figure 1: The overall pipeline of the FRL model. Given a document about the environment dynamics, a goal description, and the observation of the initial environment, the manager module will generate a plan to reach the goal in a backwards manner. The worker will fulfill the sub-goals of the plan one by one according to the observation of the environment and the sub-goal. The red arrows and the black arrows represent the data flow with and without gradient propagation respectively.

complete the goal, when they have access the game and the manual. These human rationales can be logically derived from the texts in the manuals, with necessary knowledge about the game setting. Therefore, it is feasible to have a natural language processing (NLP) model to learn the reasoning process of manual reading, by imitating the human players. However, the results from this reasoning process mainly specify classes of target states an agent should arrive at in a higher level, but do not correspond to any specific action an agent can take. As a result, there exists a gap between the reasoning process in the manual reading and the actual operations a game agent could perform.

The aforementioned semantic mismatch introduces difficult to the reading-to-act agent design: To make agent aware of the high-level textual information, existing studies (Zhong et al., 2020; Wang & Narasimhan, 2021) feed the instructions and manuals to the agent at each step. The agent policy models then rely on the cross-attention between the texts and the environment states to achieve a text-enhance state representation, from which an action is predicted. However, like the human rationales we show earlier, the strategies derived from the texts usually cannot directly map to the action space. With the existence of this semantic mismatch, the agents map textual form of knowledge to actions in an indirect and implicit way, thus are not powerful and efficient enough to handle the reading-to-act tasks.

We deal with this semantic mismatch challenge in two folds. First, we propose a feudal reinforcement learning (FRL) framework to handle the semantic mismatch. The framework consists of two agents. The **manager** agent works on the high-level abstract information from the texts, and specifies a strategy, i.e., a sequence of sub-goals, to achieve the final goal. In other words, the manager agent reads the instructions and manuals to *propose a plan*. Then another **worker** agent achieves the sub-goals in a plan one-by-one. The worker works with the low-level perceptions and actions in the environments, and is rewarded according to whether it accomplishes the sub-goals specified by the manager. Second, we equip the manager agent with a multi-hop reader model, so that it can better generate the high-level plans when the goal require multiple steps of reasoning to achieve. This enhanced manager architecture, named the **multi-hop plan generator**, aims to address the challenging scenarios where the sub-goals need to be inferred conditioned on both (1) multiple pieces of the textual knowledge and (2) the relation between the history sub-goal sequence and the final goal. Specifically, the multi-hop generator predicts the sub-goals in a backward manner – starting from the final goal, it sequentially predicts sub-goals, with each one enables its next sub-goal based on

the text knowledge. Like the example in Figure 1, with the ghost generated as a sub-goal that relates to the target army, the model should reason to get a weapon that can defeat the ghost as another sub-goal, so that completing both sub-goals fulfills the end goal.

Our FRL framework effectively alleviated the mismatch between high-level textual information and low-level perceptions and actions; and is general to various forms of environments, instructions and manuals. Our multi-hop plan generator, on the other hand, can significantly boost the FRL framework's ability to deal with challenging tasks where reasoning over the texts are critical to resolving the instructed goals. We verify our approach mainly on the challenging benchmark RTFM (Zhong et al., 2020) and verify the generalization ability on the benchmark Messenger (Wang & Narasimhan, 2021). On both benchmarks, prior methods are not able to generate non-trivial results with straight-forward end-to-end training. Some limited progresses have be made when these methods are trained with human-designed curriculum. However, the design of the curriculum makes the *assumption of white-box environments*, i.e., the curriculum requires modifications on the environment simulators to simpler versions. In comparison, our proposed approach achieves good performance with no need for curriculum learning with simulator modifications, giving a more powerful and more realistic solution. Specifically, our approach achieves near perfect results on the RTFM benchmark, leading to $> 60\%$ and $> 40\%$ absolute win rate improvement compared to the previous best numbers without and with curriculum learning, respectively.

## 2 METHODOLOGY

Feudal Reinforcement Learning (FRL) (Dayan & Hinton, 1993) proposes a managerial hierarchy that dissects a holistic task into multi-level sub-tasks to speed up the reinforcement learning process. It creates multi-level manager-workers where high-level managers set tasks for the low-level workers and, in return, the low-level workers aim to accomplish the tasks set by the high-level managers. In conjunction with deep learning (Goodfellow et al., 2016), various FRL methods (Vezhnevets et al., 2017; Nachum et al., 2018; Casanueva et al., 2018) achieve breakthroughs on specific domains such as video games, dialogue management and robotics etc.. In our case, we aim to solve a more complicated multi-domain task – our agent is required to comprehend the linguistic guidance and act properly in a visual environment to achieve a goal. Observing that the agent's actions subordinate to the language guidance, we propose a hierarchical manager-worker framework. As shown in Figure 1, the high-level manager translates the linguistic information into a series of sub-goals, and the low-level worker tries to fulfill the sub-goals by making specific actions in the visual environment.

### 2.1 TASK SETTINGS

We evaluate our model on RTFM (Zhong et al., 2020) task. In RTFM, the agent is given a document of environment dynamics, observations of the environment, and an underspecified goal instruction. The document includes information about which monsters belong to which team, which modifiers are effective against which element. The goal indicates which team the player should defeat. Both of the document and the goal are generated from human-written templates. The environment is represented as a matrix of text in which each cell describes the entity in the cell. When the player is in the same cell with an item or a monster, the player will pick up the item or engage in combat with the monster. The player can carry only one item at a time. When encountering a new item, the player will pick up the new item and lose the previous item forever. A monster moves towards the agent with $60\%$ probability, and moves randomly otherwise. In a full RTFM task, there will be two monsters and two items as the target and the distractor respectively. To accomplish the task, there are multiple reasoning and action steps to do: (1) identify the target team from the goal; (2) identify which monster in the environment belongs to the target team, and its element; (3) identify the modifiers effectively against the element of the target monster; (4) identify which item in the environment has the desired modifier; (5) pick up the target item; (6) beat the target monster. If the agent fails to carry the target item or engages a combat with the distractor monster, it will lose the game. The agent receives a reward of $+1$ if it wins and $-1$ otherwise.

Furthermore, a recent work proposed a related reading-to-act environment named *Messenger* (Wang & Narasimhan, 2021) in concurrent to our work of this paper. For completeness, we conduct experiments on the *Messenger* task and report the results in Appendix A.

## 2.2 FEDUAL REINFORCEMENT LEARNING FORMULATION

In the fedual reinforcement learning (FRL) formulation, both the manager and the worker subject to Markov Decision Processes (MDP). The two agents differ in the spaces of decision making: the manager generates sequential targets based on language inputs; the worker generates a series of control signals to drive the agent in the environment, only if they cooperate can the model attain the final goal.

Specifically, the manager strives to generate an accurate target and pass it to the worker while the worker maximizes its capability to reach that target. In our method, the states of the manager consists of the input manual, the goal instruction, and the history sub-goals, and the action of the manager is a new generated sub-goal, which is an object in the game environment. The worker agent only sees the sub-goal from the manager and the lower level game environment without the manual. Besides, the sub-goal from the manager is transformed to the position of the sub-goal object in each time step. Therefore, in each step, the state of the worker consists of the current game state plus an indicator of the sub-goal position.

**Rewards** For the worker, the problem setting is analogous to the Maze problem (Barto et al., 1989). Given the target object and the current state, the agent is expected to make actions to move toward the target. If it finally touches all the targets generated by the manager, the sequence of actions will be rewarded; if it touches a wrong object or dies, the sequence of actions will be penalized.

For the manager, the generation of the next target is based on the word description and the current target. Its RL loss is from the final reward of the game. In the game setting, we judge whether the agent successfully achieves or fails the goal instruction according to the true rewards returned by the environment. The sequence of generated sub-goals will be rewarded for the prior case and be penalized for the latter one.

We can plug in the Bellman equation (Baird, 1995) and follow a typical policy learning schedule (Zhong et al., 2020) to optimize this manager-worker system.

## 2.3 WORKER

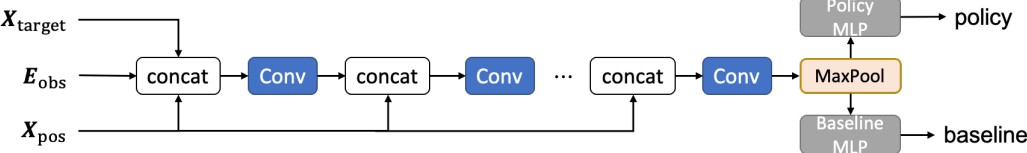

Figure 2: The structure of the worker agent.

The worker interacts with the observation from the environment and the sub-goal from the manager. In this case of a textual environment, we treat the grid of word embeddings as the visual features for the worker.

We use a residual CNN model as our backbone for the worker. The worker consists of 5 convolution layers, as shown in Figure 2. There is a residual connection from the third layer to the fifth layer. Let $\mathbf{E}_{\mathrm{obs}}$ denote word embedding corresponding to the observations from the environment, where $\mathbf{E}_{\mathrm{obs}}[:,:,i,j]$ represents the embeddings corresponding to the $l_{\mathrm{obs}}$-word string that describes the objects in location $(i,j)$ in the grid world. For each cell, the positional feature $\mathbf{X}_{\mathrm{pos}}$ consists of the $x$ and $y$ distance from the cell to the player respectively, normalized by the width and height of the grid-world. The target feature $\mathbf{X}_{\mathrm{target}}$ is the $x$ and $y$ coordinates of the sub-goal target in the grid world. The input to each layer consists of the output from the previous layer, concatenated with positional features. For the $i$th layer, we have

$$\mathbf{R}^{(i)} = [\mathbf{V}^{(i-1)}; \mathbf{X}_{\mathrm{pos}}] \tag{1}$$

$$\mathbf{V}^{(i)} = \mathrm{Conv}(\mathbf{R}^{(i)}) \tag{2}$$

For $i = 0$, we concatenate the bag-of-words embeddings with the target feature as the initial visual features $\mathbf{V}^{(0)} = [\sum_k \mathbf{E}_{\mathrm{obs},k}; \mathbf{X}_{\mathrm{target}}]$. We do $\mathbf{v} = \mathrm{MaxPool}(\mathbf{V}^{(\mathrm{last})})$ over the spacial dimension

and compute the policy $\mathbf{y}_{\text{policy}}$ and $y_{\text{baseline}}$ as

$$\mathbf{y}_{\text{policy}} = \text{MLP}_{\text{policy}}(\mathbf{v}) \tag{3}$$

$$y_{\text{baseline}} = \text{MLP}_{\text{baseline}}(\mathbf{v}) \tag{4}$$

where $\text{MLP}_{\text{policy}}$ and $\text{MLP}_{\text{baseline}}$ are 2-layer perceptrons with Tanh activation.

## 2.4 MANAGER

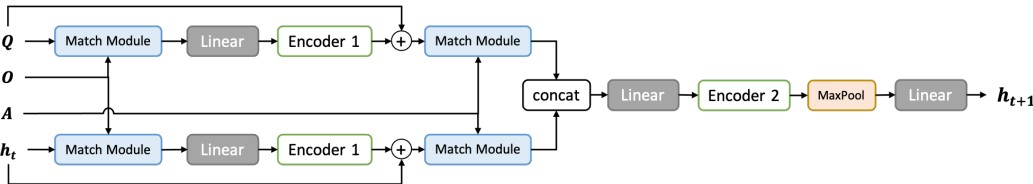

Figure 3: The structure of the manager agent.

The manager aims to predict the step-wise target object given the textual information and the agent's past targets. Inspired by human logic process in such complicated task, the reasoning order of the sub-goals is in reverse of the action order, i.e. to fulfill the goal instruction, it needs to identify the target monster; to beat the target monster, it needs to identify the target item. The overall structure of the manager is shown in Figure 3.

The inputs $(Q, O, A, H)$ to the manager are encoded textual embeddings from a shared bidirectional LSTM (Hochreiter & Schmidhuber, 1997; Schuster & Paliwal, 1997). We denote $d_e$ and $d_n$ as the embedding dimension and the length of an object name respectively. $\mathbf{Q} = \{\mathbf{q}_0, \mathbf{q}_1, \ldots, \mathbf{q}_N\} \in \mathbb{R}^{N \times d_e}$ represents the word embeddings for the goal description. We concatenate the wiki paragraph with all object names and denote their encoded embeddings as $\mathbf{O} = \{\mathbf{o}_0, \mathbf{o}_1, \ldots, \mathbf{o}_M\} \in \mathbb{R}^{M \times d_e}$. $\mathbf{A} = \{\mathbf{a}_0, \mathbf{a}_1, \ldots, \mathbf{a}_U\} \in \mathbb{R}^{U \times d_n \times d_e}$ represents the embeddings for object names. The past target object embeddings are denoted as $\mathbf{H} = \{\mathbf{h}_0, \mathbf{h}_1, \ldots, \mathbf{h}_V\} \in \mathbb{R}^{V \times d_n \times d_e}$.

The manager architecture derives from a Co-match LSTM model (Wang et al., 2018b). The Co-match LSTM was first proposed for matching among multiple sequences, e.g., matching a question and an answer option to a paragraph in Wang et al. (2018b). Wang et al. (2019a) applies this model to multi-hop reasoning over texts, benefiting from its ability to capture sequence order information. In our case, four information sources need be matched with each other and thus the original Co-match LSTM (Wang et al., 2018b; 2019a) cannot be simply plugged in. Borrowing the idea of matching sequences, we design a match module $\mathcal{M}(X, Y)$ that performs the following operations:

$$
\begin{aligned}
e_{jk} &= \vec{x_j}^T \vec{y_k} \\
\tilde{x}_j &= \sum_{k=0}^{Z} \frac{\exp(e_{jk})}{\sum_{l=0}^{Z} \exp(e_{jl})} \vec{y_k} \\
\tilde{m}_j &= [x_j, \tilde{x}_j - x_j, \tilde{x}_j, x_j * \tilde{x}_j] \\
\tilde{M} &= [\tilde{m}_0, \tilde{m}_1, ..., \tilde{m}_W],
\end{aligned}
\tag{5}
$$

where $X = \{\vec{x_0}, \vec{x_1}, ..., \vec{x_W}\} \in \mathbb{R}^{W \times d_e}$ and $Y = \{\vec{y_0}, \vec{y_1}, ..., \vec{y_Z}\} \in \mathbb{R}^{Z \times d_e}$.

Our model needs predict the next target object based on the current attained target (current state) while bearing the final goal in memory. For example, in Figure 1, after the agent acquires the arcane cutlass, our model should acknowledge that the agent has that weapon in hand and its goal is to defeat the Star Alliance. The next target prediction should subject to the constraints of Wiki and be one of the observed objects. Therefore, we design a dual-path framework where one path projects the constraint of wiki ($O$) to the the current state ($h_t$) and projects the result ($h_t^O$) to the observed objects ($A$); the other path projects the constraint of wiki ($O$) to the final goal ($Q$) and projects the results ($Q^O$) to the observed objects ($A$). Then, we concatenate the results of these two paths and regress the concatenated features to probabilities of all observed objects. Finally, we select the object with the highest probability as the next target.

The specific design of our model is shown in Figure 3. Encoder 1 ($\mathcal{E}_1$) and Encoder 2 ($\mathcal{E}_2$) are two different bidirectional LSTMs. Linear represents independent multi-layer perceptrons (MLP) and is denoted by $\mathcal{F}^i$. We further denote $\mathcal{C}$ as the concatenation operation, $\mathcal{B}$ as the maxpool operation, $\mathcal{P}$ as the expand operation and $\mathcal{R}$ as the reshape operation. We describe the detailed operations of our manager with Eqn. (6).

$$
\begin{aligned}
Q^O &= [\mathcal{E}_1(\mathcal{F}^1(\mathcal{M}(Q,O))) + Q] \in \mathbb{R}^{N \times d_e}; \overline{Q^O} = \mathcal{P}([Q^O, ..., Q^O]) \in \mathbb{R}^{U \times N \times d_e}; \\
h_t^O &= [\mathcal{E}_1(\mathcal{F}^2(\mathcal{M}(h_t,O))) + h_t] \in \mathbb{R}^{d_n \times d_e}; \overline{h_t^O} = \mathcal{P}([h_t^O, ..., h_t^O]) \in \mathbb{R}^{U \times d_n \times d_e}; \\
A^{QO} &= \mathcal{M}(A, \overline{Q^O}) \in \mathbb{R}^{U \times d_n \times 4d_e}; A^{hO} = \mathcal{M}(A, \overline{h_t^O}) \in \mathbb{R}^{U \times d_n \times 4d_e}; \\
A^{QOh} &= \mathcal{C}([A^{QO}, A^{hO}]) \in \mathbb{R}^{U \times d_n \times 2d_e}; \tilde{A}^{\alpha} = \mathcal{F}^3(A^{QOh}) \in \mathbb{R}^{U \times d_n \times d_e}; \\
\tilde{A}^{\beta} &= \mathcal{B}(\mathcal{E}_2(\tilde{A}^{\alpha})) \in \mathbb{R}^{U \times d_e}; \tilde{A} = \mathcal{R}(\mathcal{F}_4(\tilde{A}^{\beta})) \in \mathbb{R}^U; x = \arg\max_i \tilde{a}_i; h_{t+1} = a_x
\end{aligned} \quad (6)
$$

## 3 EXPERIMENT

### 3.1 TRAINING DETAILS

The aforementioned reasoning and action steps in Section 2.1 can be divided into a reasoning part and a action part evidently. The manager only needs to learn to identify the target monster and the target item sequentially, while the worker only needs to learn to get to the target object and avoid all other objects with the sub-goal given by the manager. In other words, we can train the manager and the worker individually as they take care of different parts of the whole task asynchronously. Since the whole environment is represented as text cells and is visible to the agent, we slightly change the environment such that all the objects(monsters and items) and their positions are a part of the observation. We shuffle the order of all objects to avoid that the agent takes advantage of order information. The only difference is that we remit the messy procedure to go through all cells to get all the items and their corresponding positions.

To train the worker, a random goal object is selected for the worker to reach. The worker needs learn to reach the goal and avoid touching any other objects to stay alive. Since the agent will die when reaching the goal if the goal is a monster. When a monster is selected as the goal, we weaken the goal monster so that the agent will not die for reaching the goal. The agent will still die if it touches the other unselected monster. The worker needs to reach the goal object and stay alive to win. We train the worker with TorchBeast (Küttler et al., 2019), an implementation of IMPALA (Espeholt et al., 2018). We use 20 actors and a batch size of 24. We set the maximum unroll length as 80 frames. Each episode lasts for a maximum of 1000 frames. The worker is optimized using RMSProp (Tieleman & Hinton, 2017) with $\alpha = 0.99$ and $\epsilon = 0.01$. It takes less than 10 hours to train the worker for 50 million frames on 1 Nvidia RTX2080ti GPU.

To train the manager, we need collect a bunch of trajectories and corresponding end-game rewards for policy update. Notice that manager only cares about the sub-goal sequence and the end-game reward, so with a well trained worker, the training of the manager degenerate to a supervised manner: if the manager generates a correct sub-goal sequence, i.e. the target monster and the target item, it gets reward $+1$, otherwise it gets reward $-1$. For each game, we can easily extract the ground truth sub-goal sequence, so we randomly generate 100 thousand initial observations and corresponding sub-goal sequences as the dataset. We split the datset as 80/20 thousand train/val set. There are more than 2 million different monster-team-modifier-element combinations without considering the natural language templates, and 200 million otherwise. Thus the sub-goal dataset is a very small part of the possible scenarios. The manager will do a 2-step reasoning to determine the target monster and target item sequentially. In the first step, we use $\langle$NULL$\rangle$ as the previous sub-goal input. In the second step, we use the firstly generated sub-goal as the previous sub-goal input. We update the parameters with cross entropy loss at each step. We use Adam optimizer (Kingma & Ba, 2015) with learning rate $10^{-4}$.We train the manager module on 1 Nvidia RTX2080ti GPU with batch size 200 for 100 epochs.

Table 1: Win rate performance on full RTFM

| Method | $6 \times 6$ | $10 \times 10$ |
|---|---|---|
| txt2$\pi$ | $23 \pm 2$ | – |
| worker (random) | $12 \pm 2.6$ | $12 \pm 0.1$ |
| FRL (forwards) | $43 \pm 0.6$ | $47 \pm 2.2$ |
| FRL (backwards) | $\mathbf{84.2 \pm 0.3}$ | $\mathbf{95.7 \pm 0.1}$ |
| txt2$\pi$ (w/ curriculum) | $55 \pm 22$ | $43 \pm 13/55 \pm 27$ [a] |
| Upperbound | $\sim 86$ | $\sim 96$ |

[a]Result generalised from model trained on $6 \times 6$ games

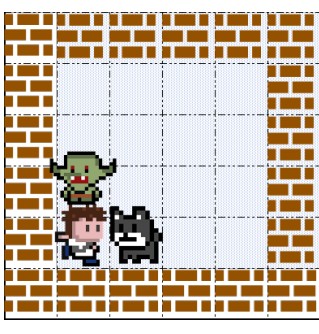

Figure 4: Common failure case in FRL

## 3.2 RESULTS AND COMPARISON

**Overall results**  The performance of our model with other models is show in Table 1. In the table, worker (random) denotes a worker with a random manager, and FRL (backwards) denotes our framework with a manager generating sub-goals in a backwards manner, i.e., with the multi-hop manager model in 2.4. FRL (forwards) is an ablation of our solution, with the manager generating sub-goals in a forward manner. We run 5 randomly initialized worker training on $6 \times 6$ and $10 \times 10$ grid-sized RTFM games respectively. Upperbound is the performance of our worker with the groundtruth sub-goals provided.

We make the following observations. First, without curriculum learning, the existing txt2$\pi$ model cannot learn policy effectively, while our model reaches near upperbound performance in both $6 \times 6$ and $10 \times 10$ RTFM games. Second, the worker (random) represents a lowerbound of our FRL framework. It achieves a win rate about $12\%$ instead of $1/4 \cdot 1/3 = 1/12$ since there is another win trajectory that the player can pick up the distractor item first, then the target item and finally beat the target monster. Still, both of our FRL models achieve much better results compared to this lowerbound. Third, txt2$\pi$ trained in $6 \times 6$ environment performs better than that in $10 \times 10$ environment. While our FRL model performs better in open environment, since the main case that causes failure is when the monsters surround the player at the corner, as shown in Figure 4. Thus in open space, the player is less likely to be trapped by monsters.

Finally, the FRL (forward) outperforms the txt2$\pi$ baseline, but is significantly worse compared to our FRL (backward) solution. In a reasoning process from the goal to specific steps, the agent would have to know all the following steps before it outputs the first step in a forwards manner. This explains why the forwards manager performs much worse than the backwards one. The results demonstrate the advantage of our multi-hop reasoning model as the manager agent. Also we can see that FRL (backward) performs similarly to the upperbound, which indicates that the manager does a near perfect job. However this may not hold in the real-world applications with noisy texts and long reasoning sequences. In these scenarios, there can be error propagation in the sub-goals generated by the manager. In future work, this should be addressed by injecting the real-time feedback from the game environment to revise the sub-goals, like in Vezhnevets et al. (2017).

At the beginning of the game, the manager will set the target item as the goal (bounded in red frame), then the worker will control the agent to reach the goal. Once the agent reaches the target item, the goal will change to the target monster. The worker continues to reach the goal and wins the game.

**Evaluation of the worker agent**  We group the log points in every 10,000-frame interval and show the worker training process in Figure 5. We take log points in the interval with upper bound $5 \times 10^7$ as the training accuracy. We run tests on all these five trained workers as testing accuracy, which is the upper bound performance our manager-worker framework can reach in theory. Since the sub-goals generated from the ideal manager are in the same latent space as the observation information, our training process is stable and converges quickly. In other words, by generating subtly designed sub-goals, the manager is able to transfer the challenging reading to act problem into some simple problems and greatly alleviate the mismatch between the high-level linguistic information and the

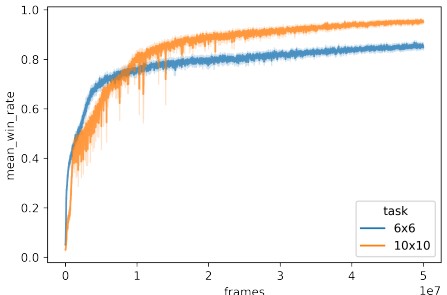

| Worker | Train Acc | Test Acc |
|--------|-----------|----------|
| $6 \times 6$ | $85 \pm 0.1$ | $85 \pm 1.7$ |
| $10 \times 10$ | $95 \pm 0.1$ | $96 \pm 0.3$ |

Table 2: Average train/test accuracy of five randomly initialized worker models.

Figure 5: Average win rate of 5 worker training runs.

low-level perception and actions. As a result, the worker can perform very well even with simple structures.

**Study of the training strategy of workers** We also compare the different reward setting for training the worker. Since in the RTFM setting, the game does not end immediately when the agent kills the monster. Instead, the system will determine whether the game ends after every round of movements for all entities. One possible training process can be set as rewarding the agent as long as it reaches the target object(item or monster) without alive requirement, and a training trajectory is done as it reaches the target object or dies. Since the agent moves first in each round, the nearby distractor monster may still kill the agent after agent kills the target monster. Another training process can be rewarding the agent for reaching a sub-goal pair generated by a perfect manager and staying alive. Compared with rewarding for reaching a single goal and staying alive, the absence of alive requirement makes the agent short-sighted during testing, as shown in Figure 6. The item bounded in the red frame is the current target item. There is a monster next to the target item, thus there is some risk that the monster reaches where the target item is, leading to the death of the agent. For the worker trained with rewarding for reaching one object without alive requirement, the worker tends to control the agent to reach the target without considering the risk. While for the worker trained with rewarding for staying alive and reaching all objects (single or pair), it tends to wait until the monster leaves the target item and avoid the risk of being killed during reaching the target. The performance for the one object rewarding worker without alive requirement is about $66\%$ and $76\%$ in $6 \times 6$ and $10 \times 10$ RTFM games respectively, which are obviously worse than the all object rewarding one shown in Table 5. The performance for the pair of objects rewarding worker with alive requirement is about $85\%$ and $96\%$ in $6 \times 6$ and $10 \times 10$ RTFM games respectively, which is close to the upperbound.

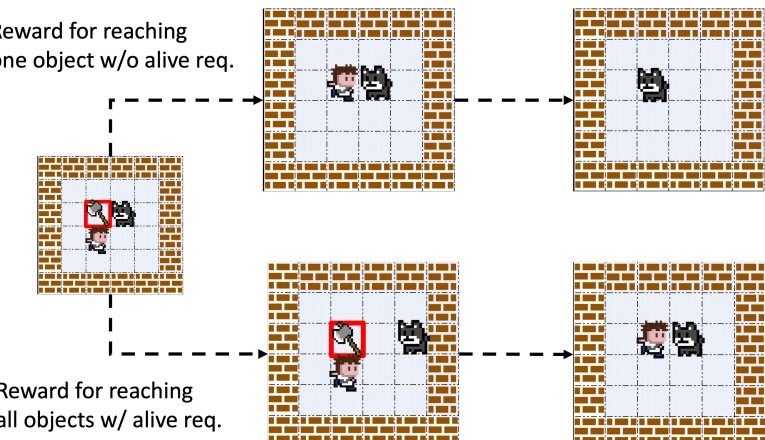

Figure 6: Worker trained with reward for reaching an object w/o alive requirement tends to be short-sighted. While the one trained with alive requirement tries to avoid risk.

## 4 RELATED WORK

**Language-conditioned reinforcement learning**  Reinforcement learning has been applied to many environments with textual observations. Representative research directions under this scope include: (1) reinforcement learning for textual instruction following; (2) reinforcement learning with textual knowledge enhancement, i.e., the reading to action direction this paper studies; (3) reinforcement learning in textual environments.

Most of the language-conditioned RL work belongs to the instruction following direction, such as visual-language navigation (Anderson et al., 2018; Wang et al., 2019b), video gaming (Hermann et al., 2017; Bahdanau et al., 201), robot control (Tellex et al., 2020) and more (Branavan et al., 2009). The language instructions in these tasks serve as a guidance to supervise the model to work in a non-linguistic domain. The instructions are usually long and concrete descriptions of action sequences.

The second direction of reading to act (Branavan et al., 2009; Narasimhan et al., 2017; Wang & Narasimhan, 2021; Zhong et al., 2020), as discussed in Section 1, differs from the conventional instruction following in two perspectives. First, the language instructions are usually short and only describe the target goal states. Second, the agents are usually provided additional text descriptions about the environment as prior knowledge. Therefore, the agents need to comprehend the text knowledge in order to derive a solution to the specific goal. In most of the example works, the text descriptions have the form of manuals of the environments.

Finally, in the natural language processing community, there are many pure text environments established for reinforcement learning research. The most important example is the dialog systems (Dhingra et al., 2017). The text games (Côté et al., 2018; Hausknecht et al., 2020) are a recent popular field in this direction. Other examples include studies that apply reinforcement learning to conventional NLP tasks, like information extraction (Narasimhan et al., 2016) and open-domain question answering (Wang et al., 2018a).

**Evaluation of machine comprehension**  Finally, we would like to point out that our work is related to a long line of work on evaluation of machine reading comprehension. Machine comprehension capability is usually evaluated as question answering (QA) tasks, from the early attempts like MCTest (Richardson et al., 2013), to many QA tasks for neural reading models like CNN/Dailymail (Hermann et al., 2015) and SQuAD (Rajpurkar et al., 2016), until some recent tasks that require deep story comprehension understanding (Kočiskỳ et al., 2018) or commonsense reasoning (Huang et al., 2019; Wang et al., 2021) skills. However, researchers also identified shortcuts for models to solve the QA tasks, questioning the appropriateness of QA as an evaluation of machine comprehension.

To deal with such deficiency, new evaluation benchmarks are proposed. A common design guidance of these tasks is to require the model to first comprehend the texts and use the comprehended facts to achieve the target goals. One example of these efforts is the multi-hop reasoning tasks (Welbl et al., 2018; Khot et al., 2020). As an alternative, recent works also studied direct evaluation of machine comprehension in interactive fiction game playing (Guo et al., 2020; Hausknecht et al., 2020; Yao et al., 2021). Our studied tasks can be viewed as the intersection between the aforementioned two types of evaluation tasks.

## 5 CONCLUSION

In this paper, we propose a Feudal Reinforcement Learning (FRL) framework to attack the challenging read-to-act problem. We design a high-level manager agent to reason and translate the multi-hop linguistic information into multi-step sub-tasks and introduce a low-level worker agent to perceive and act in the environment to achieve the tasks set by the manager. Our framework effectively solve the mismatching problem between the text-level inference and the low-level perception and action without human-designed curriculum. We conduct experiments on challenging tasks including Read to Fight Monsters (RTFM) and Messenger. We analyze the module functions with adequate ablation studies and show that our model achieves a far better performance that those of state-of-the-art models. To our best knowledge, we do not identify significant negative impacts on society resulting from this work.

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

# A ADDITIONAL EXPERIMENTS IN THE MESSENGER ENVIRONMENT

## A.1 SETTINGS

**Task**  We also evaluate our model on *Messenger* (Wang & Narasimhan, 2021) stage 2 and stage 3 tasks.

In Messenger stage 2, there are three different entities: an enemy, a message, and a goal. Each of the entities is assigned a *stationary, chasing*, or *fleeing* movement type. The player is provided with the manual of descriptions for each of the entities. The objective of the player is to bring the message to the goal while avoiding the enemy. If the player encounters the enemy during the game, or the goal without obtaining the message, it loses the game and obtains a reward of $-1$. Rewards of $0.5$ and $1$ are provided for obtaining and delivering the message to the goal respectively.

In Messenger stage 3, a distractor message and a distractor goal are added. These two distractor entities have the same names as the goal entities but different motion types. The agent cannot distinguish the same named entities without observing their motion during the game. Other game settings are the same as the stage 2.

Different from the RTFM task, the agent needs to learn the one-to-one mapping from each of the description sentences to each of the entities in the visual space. There are not multiple reasoning steps to do to accomplish the task.

**Training details**  Similar to training on the RTFM, we train the worker and the manager separately.

To train the manager, because of the gap among train, val, and test environments in Messenger, we randomly generate 1000/500/500 initial observations from train/val/test environments of stage 1 and stage 2 respectively. Since there is no explicit "task sentence" in the Messenger environment, we set "get message then get goal" as the task sentence for the manager. In stage 2, the manual is enough for determine the one-to-one mapping. While in stage 3, since the manager works at the beginning of each game, it cannot distinguish entities with the same name. So in both cases, only the manual without the entities is encoded as **O**. There is no difference between forward and backward manager, because the agent does not need the previous target to reason the next target, in other words, it only need to learn the one-to-one mapping. We use the same cross entropy loss and Adam optimizer (Kingma & Ba, 2015) with learning rate $10^{-4}$. We train the manager module on 1 Nvidia RTX2080ti GPU with batch size 100 for 100 epochs.

To train the worker, we follow the same method as the RTFM task. We use TorchBeast (Küttler et al., 2019) with 20 actors and batch size of 24. We set the maximum unroll length as 80 frames and RMSProp (Tieleman & Hinton, 2017) as the optimizer.

## A.2 RESULTS AND COMPARISON

We show the worker performance in Stage 2 with ideal manager sub-goals in Table 3, which can be considered as the upperbound for our FRL model. Given correct sub-goals, our worker agent can accomplish the task in a near perfect performance.

We perform evaluation in the Messenger test environment, which includes previously unseen games. As shown in Table 4, EMMA (Wang & Narasimhan, 2021) cannot learn anything without curricu-

Table 3: Average worker win rate in training, validation, and testing environment of Stage 2.

| Train Win Rate | Val Win Rate | Test Win Rate |
|---|---|---|
| $99.5 \pm 0.2$ | $99.7 \pm 0.1$ | $94.6 \pm 0.9$ |

lum. Although Wang & Narasimhan (2021) claim the player need to observe the dynamics to learn the mapping between the description sentences and the entities, they set a special Stage 1 during which all entities are immovable. In such case, there is no dynamic difference between different entities. And the specific character representing the entity in the visual space has a specific meaning, such as mage, queen and so on. Also shown as the win rate of Stage 3 of EMMA, the huge

Table 4: Performance on Messenger

| Method | Stage 2 Test Win Rate | Stage 3 Test Win Rate |
|---|---|---|
| EMMA | – | – |
| FRL | $\mathbf{93 \pm 0.2}$ | $\mathbf{13 \pm 1.4}$ |
| EMMA (w/ curriculum) | $85 \pm 0.6$ | $10 \pm 0.8$ |
| Upperbound | $\sim 95$ | $\sim 95$ |

drop of performance indicates that their model is actually learning the mapping from the description sentences to the characters in Stage 1 instead of learning the mapping through reasoning and interacting with the environment. While our manager agent well connects the linguistic information and the visual information, our model achieves extraordinary performance in Stage 2. However, due to the indistinguishable setting for the entities, the performance of our model also drops. With the initial observation, which has no motion information, the manager cannot distinguish the same named entities, but randomly guess the goal entity between them. We argue it is not a suitable setting for *reading-to-act* task, because with only the manual description, it cannot provide all the information to finish a task.

