# OpenReview forum: "Feudal Reinforcement Learning by Reading Manuals"
_ICLR.cc/2022/Conference — ICLR 2022 Submitted_

### Official Review · Reviewer_tKa7 · 2021-11-02

**Correctness:** 3
**Technical Novelty And Significance:** 2
**Empirical Novelty And Significance:** 3
**Recommendation:** 5
**Confidence:** 2

**Main Review:**

Strengths
- I find the application of Feudal Reinforcement Learning (FRL) interesting and novel in this setting.

Weaknesses
- the paper is quite hard to follow and it misses many details. On page 4, the authors mention the Bellman equation and "a typical policy learning schedule". The authors do not provide any formal definition of which exact RL method has been used, e.g. on/off policy etc. Moreover, it is very hard to read/understand Eq. 1. and 2. I suggest the authors rewrite it and better connect them to the figures.
- [EDIT] Clarified in the rebuttal [EDIT]. The results reported in Table 1 doesn't match the one from txt2$\pi$ (Zhong et al. 2020), which achieve 84±21 in the training environments,  83±21 in the 6x6 and 66±22 in 10x10. Could the authors' comment on the results in Table 1? is the setting comparable?
- the paper claims, in the introduction, that the evaluation is done in two environments. However, the experiments in Messager are only preliminary and only shown in the appendix. I suggest explicitly mentioning this in the intro.

**Summary Of The Paper:**

In this paper, the authors propose a Feudal Reinforcement Learning (FRL) algorithm for improving the miss-match between high-level natural language commands (e.g., "share my recent photo to my parents") and the actual complex set of operations required by the systems (e.g., open the album --> select recent photo --> etc.). FRL (Dayan & Hinton, 1993) is a hierarchical RL algorithm with two agents: manager and worker. The manager issues a plan which is executed by the worker. To elaborate, the manager learns to generate a sequence of targets given $Q$ the goal description, $O$ the wiki paragraph which describes the game, $A$ the object names in the environment, and $H$ the subgoal history ($h_t$ refers to the target object at time $t$). The worker, instead, uses the plan from the manager (in terms of coordinate to the target ($X_target$)), the observation $E_{obs}$, and the position to the other player ($X_{pos}$) (not sure I understood correctly this) to interact with the environment.

The manager and the worker are trained separately using imitation learning (also here it is a bit confusing), from a 100K trajectory collected by a random walk in the map.

The authors benchmarked the FRL with two text-based interactive games such as RTFM (Zhong et al. 2020) and Messager (Wang & Narasimhan, et al 2021) (the results are only in the appendix, but this paper is very recent). The results show that FRL is better than txt2$\pi$ without curriculum learning (here I am a bit unsure because the results in the original paper are different), in the RTFM.


**Summary Of The Review:**

An interesting application of Feudal Reinforcement Learning (FRL) in text-guided RL environments (e.g., RTFM (Zhong et al. 2020)). Lack of details limits the significance of the paper.

---

> ### Author Response · Authors · 2021-11-13
> **response to Reviewer tKa7**
>
>
> >> RL method
>
> Similar to $txt2\pi$ (Zhong et al. 2020), we use IMPALA (Espeholt et al., 2018) to train the worker. When unrolling actors, we use a maximum unroll length of 80 frames. Each episode lasts for a maximum of 1000 frames. We optimise using RMSProp (Tieleman & Hinton, 2012) with a learning rate of 0.005, which is annealed linearly for 100 million frames. We set $\alpha = 0.99, \epsilon = 0.01$. We mention the training method in training details.
>
> We will add more details of the training in the appendix.
>
> >> Results difference of $txt2\pi$
>
> We follow the exact setting as $txt2\pi$(Zhong et al. 2020). There are 2 different environments of RTFM, train and eval(test). According to their paper, "No assignments of monster-team-modiﬁer-element are shared between train and eval to test whether the agent is able to generalise to new environments with dynamics not seen during training via reading". The $83 \pm 21$ in $6 \times 6$ and $66 \pm 22$ in $10 \times 10$ are the final performance of the $txt2\pi$ model in the train environment with curriculum learning. The data we report in our paper is the performance they claimed in the eval environment. It's in Table 3 of $txt2\pi$(Zhong et al. 2020). We follow the train and eval setting exactly the same as $txt2\pi$, i.e. training in the train environment and testing in the eval environment. There is almost no drop of performance between our overall model and the worker part, which indicates our manager learns to perfectly reason the subgoals by reading the manual during the training.
>
> >> Messenger result in the appendix
>
> We have to point out that the Messenger is not a very good scenario for reading-to-act tasks. In Messenger, the entities are represented as a single character in the environment, while with several synonyms in the manual set. The manual describes the dynamic of each entity. Since the manual sentence is one to one mapped to the entity, there is no reasoning step needed. We test the performance in Messenger mainly to show the generalization ability of our model. We will explicitly mention that the Messenger result is in the appendix in our paper.

---

> > ### Comment · Reviewer_tKa7 · 2021-11-28
> > **Re: Response**
> >
> > Thanks for the clarification.

---

### Official Review · Reviewer_eXEz · 2021-11-02

**Correctness:** 3
**Technical Novelty And Significance:** 3
**Empirical Novelty And Significance:** 3
**Recommendation:** 6
**Confidence:** 4

**Main Review:**

The paper is clearly written and easy to understand. The authors motivate the problem and introduce their model in relation to past works quite well.

**Clarity in Model Specs:** While describing the model, the Co-match LSTM model. I found this part to be unclear. I would urge the authors to introduce the intuitions and notations used in equations (1) before presenting them to make that section clearer. Further, Figure 2 shows linear layers, but the authors say that "Linear represents independent MLPs". I then urge the authors to change the figure.

**Collecting data for training the manager:** The authors state that the manager is trained by letting an agent walk randomly to generate a trajectory and using the successful trajectories as supervision for the manager. This seems like an inefficient way to collect successful trajectories by letting an agent execute random actions. I would have liked to see more details about how many random trajectories were needed before sufficient successful trajectories were collected. Moreover, collecting successful trajectories is not possible in more complex environments. I urge the authors to discuss how their manager model would learn in such scenarios.

**Cases of failure of FRL:** The authors do an excellent job studying ablations and how different training strategies affect the policy. I would have liked to see the failure cases of the model and why the authors think that the model fails on these scenarios.

**Results on Messenger:** The authors include the results of messenger in the appendix. The results are pretty good but are barely included in the main text. I would have liked to see some of them discussed (at least broadly, without specifics; I understand that there are space constraints). Moreover, I would have liked to see a description of the Messenger benchmark with a figure (like that of RTFM) and a description of the different splits used to evaluate (in the appendix). Finally, as the model fails on Stage 3 of Messenger, I would have liked to see a few failure cases and why the authors think the model fails, along with a speculative discussion of how it could be mitigated.

Typos:

Page 4: Our model need → Our model needs

**Summary Of The Paper:**

The paper presents a hierarchical RL model to solve tasks that require reading and understanding instructions or manuals to solve complex goals (these goals usually require multiple steps of solving subgoals). The proposed model has two parts: a manager that uses the instructions/ manuals to design subgoals, and a worker that solves the subgoals and deals with low-level perceptions. The manager and the workers are trained separately. The Feudal model achieves SOTA scores on Messenger and RTFM without the use of a hand-designed curriculum.

**Summary Of The Review:**

The FRL model shows impressive results on Messenger and RTFM. Still, there are a few improvements that the authors can make to improve the clarity, the analysis of results and also provide a more detailed discussion about cases of failure, especially on Messenger. One weakness of the model is how the data is sampled to train the manager, additional discussion on how this can be scaled to more complex environments is needed.

---

> ### Author Response · Authors · 2021-11-13
> **response to Reviewer eXEz**
>
> >> Clarity in Model Specs
>
> We appreciate the suggestion of introducing the intuitions and notations before eq(1). The intuition is that to handle the concise instructions in reading-to-act tasks, the reasoning process should follow the logic process as humans, which is in the backward manner of actual actions need to be done. We will refine the model specs in the future edition.
>
> >> Collecting data for training the manager
>
> To make it clear, we would like to point out that the training of the manager does not depend on the whole trajectory but objects the agent visits during the trajectory. With a worker trained to reach a given goal, the manager generates a pair of sub-goals and receives the corresponding final rewards to train. In other words, the training of the manager will degenerate to a supervised training process, which requires the sequence of correct sub-goals. So in such a manner, we can easily generate a bunch of training data for the manager by selecting an object sequence in the environment without letting the agent really walk around. So in a more complicated environment, we can also efficiently collect data to train the manager.
>
> We will refine the training details of the manager in the paper.
>
>
>
> >> Cases of failure of FRL
>
> In RTFM, except for setting the wrong sub-goals, the only failure scenario we observe is shown in Figure 4, where the agent is trapped at the corner by two monsters. This also conforms to the performance increase from 6x6 environment to 10x10 environment, since the agent is less likely to be trapped in a more open environment. Also, this indicates the drawback of the txt2pi model, since it performs worse in a more open environment. It shows that the txt2pi model does not fully tackle the RTFM task.
>
> >> Results on Messenger
>
> We have to point out that the Messenger is not a very good senario for reading-to-act tasks. In Messenger, the entities are represented as a single character in the environment, while with several synonyms in the manual set. The manual describes the dynamic of each entity. Since the manual sentence is one to one mapped to the entity, there is no reasoning step needed. We argue that their EMMA model learns the correspondence between the character and the synonyms, since it needs stage 1 as curriculum learning, in which all entities are fixed, i.e. no dynamic difference among them. And without stage 1, their model cannot get a reasonable performance.
>
> In Messenger stage 2, similar to RTFM, the main failure case is the agent being trapped. While in stage 3, Messenger adds distractor entities with same character representations but different dynamics. In such a case, our model cannot distinguish the distractor target and the goal target by just reading the manual and observing the environment at the game start. It has to guess between two entities with the same appearance. We believe this is also the main reason for the performance drop of EMMA. And this performance drop also indicates EMMA only learns the one to one mapping between entity characters and synonyms instead of distinguishing by dynamics (as they claimed).
>
> We will add more details and ablation study of Messenger in our paper.

---

> > ### Comment · Reviewer_eXEz · 2021-11-27
> > **Re: response to Reviewer eXEz**
> >
> > Thank you for your response!
> >
> > **Training the Manager**: Oh I see, so the training is happening at the object level subgoals; this would significantly reduce the complexity and make the random sampling more feasible. But this does introduce the prior of object-based goals and plans.
> >
> > **Messenger**: Thank you for clarifying the details for messenger! Although, I couldn't see some of the changes in the revision.
> >
> > **Clarity**: I still think that the paper lack clarity in several places, even in the revision.
> >
> > For now, my score remains the same.

---

### Official Review · Reviewer_RC7c · 2021-11-03

**Correctness:** 3
**Technical Novelty And Significance:** 2
**Empirical Novelty And Significance:** 4
**Recommendation:** 3
**Confidence:** 5

**Main Review:**

Positive points:

1. There seems to be quite a large gain in performance with the FRL model that generates sub-goals in a backwards manner.
2. Although, this seems trivial, especially given that the "upperbound" i.e., a model that has perfect subgoal information gets nearly the same performance as the FRL-backward model.

Negative points:

1. There are several things in the paper that are not clearly explained at all that made this hard to follow. Example sentences are:
a) "It achieves a win rate about 12% instead of 1/4 · 1/3 = 1/12"
b) "In the fedual reinforcement learning (FRL) formulation, both the manager and the worker subject to Markov Decision Processes (MDP)."
2. There doesn't seem to be significant/novel introduction in terms of modelling changes for the model.

**Summary Of The Paper:**

This paper proposes a feudal reinforcement learning model to solve the task of reasoning over textual (language) instructions that incorporate with low level state/action/environment information. This feudal model is composed of two agents, one of which is a "manager" that generates a multi-hop plan composed of several subgoals, while the other is a "worker" that takes in low-level information and executes actions to solve each sub-goal. They empirically evaluate on two challenging domains and show competitive performance to baselines.

**Summary Of The Review:**

Limited novelty and improvements but significant gains in performance---although unclear because the writing isn't clear.

---

> ### Author Response · Authors · 2021-11-13
> **response to Reviewer RC7c**
>
>
> >> Clarity
>
> It seems that there is some misunderstanding of the RTFM task. We have mentioned the reason why "It achieves a win rate about 12% instead of 1/4 · 1/3 = 1/12" after the corresponding sentence. There are two possible win trajectories since the agent can pick up the distractor item and still win as long as it follows a correct routine afterwards.
>
> Besides, the upper bound is not trivial. The upper bound is the theoretically best performance our model can achieve. It's not "an arbitrary model that has perfect subgoal information gets nearly the same performance as the FRL-backward model". The performance comparison between the upper bound and our model shows that our model can handle the high-level concise instructions almost perfectly by reading the manual.
>
> >> Novelty
>
> The novelty of our work is twofold: (1) We formulate the reading-to-act task as reading comprehension, hence we have the FRL framework. Also this formulation allows us to borrow the advanced reading comprehension models developed by the QA community so as to set up the new performance standard of this challenge. (2) We further formulate the sub-goal prediction as multi-hop QA and propose a novel manager model.
>
> We are not sure what the reviewer means by "modeling changes". Compared with txt2pi, our model applies a totally different hierarchical framework that does not need any human designed curriculum and achieves near perfect performance.

---

### Official Review · Reviewer_KMX2 · 2021-11-04

**Correctness:** 4
**Technical Novelty And Significance:** 3
**Empirical Novelty And Significance:** 3
**Recommendation:** 5
**Confidence:** 4

**Main Review:**

Using the Neurips review:

Originality: The key novelty I find is to use a heirarchical RL approach to segregate the transformation of the textual instructions into goals and the actual RL style execution of the goals. This is a natural approach and intuitively might be generalizable to other contexts. The design of the manager using the backward multi-hop generator is somewhat novel. Experiments are mostly straightforward though  there is one  minor novelty in the ablation to a forward generating manager.

Quality: I like the basic idea of the approach, i think a well-executed version would be worthy of an ICLR paper. But there are some concerns about what is done so far in this paper:
a. The modeling of the language seems out of date, using LSTMs etc. Transformer-based would be more state of the art (and really easy to implement I imagine using off the shelf tools these days), and the cross-attention mechanism would be an elegant way to do what the (hacky-looking?) matching module seems to be doing.

b. Experiments seem hyper-specific to the RTFM environment. This is really concerning for being able to generalize conclusions of this work to other (esp. real world) applications. For example, I see it quite possible that the results of different modeling approaches are dependent on the way the instructions are written. For a different set of instructions, forward generation by the manager might be better (e.g if the instructions were more like a step-by-step plan, in which case forward would be better than reverse because it would help resolve anaphora better). Many of the discussion of the results seem too fixated on the details of this particular environment (e.g. the "Study of the training strategy", where a lot of discussion is devoted to explaining interactions of player and monster etc, how would we draw insights for other problems from this?).

To be fair though, the experimental results do seem impressive in the sense of the delta vs baseline (I am charitably assuming text2\pi is a competitive sota model from the literature, the authors dont mention anything about it). In fact, for these environments, the problem is practically solved, which is notable for a challenge problem recently published.

Clarity: This is the biggest weakness. There are numerous grammatical and semantic mistakes, that make it impossible to understand the author's intent or requires multiple readings and putting peices in different places together to understand what's going on. Sec 2.2 took a while, part of the problem is that the problem setting was not described rigorously, so I had to figure out by implication what a "object name"  is and why we need to add d_n extra dimensions, what is meant by "grid of word embeddings" etc. I dont really understand what the reasoning for the precise operations in the  match module defined in eq 1 are, some explanation by the author could have been useful.  On the flip side, a lot of space is devoted to details that can be moved to the appendix like learning rates and an unnecessary subsection on machine comprehension related work (for example). This would have freed up space for moving the Messenger experiments into the main body, which would allow for a more comparative analysis of experiments leading to more generally applicable conclusions. A lot of work is needed to get the paper into a readable shape, and we can't trust that it will be done based on the current draft alone.


**Summary Of The Paper:**

The paper applies a feudal RL algorithm to solve text-adventure type problems where information is available in text form about the environment. A manager agent reads the text and generates a sequence of subgoals, and worker agents execute the subgoals to reach the final goal. The models are described, experiments are described (mostly in the RTFM but a callout to Messenger is there in Appendix).

**Summary Of The Review:**

A nice natural idea for doing RL with access to text instructions, experiments on RTFM impressive but only 1 setting is a negative. Poor readability is the biggest negative.

---

> ### Author Response · Authors · 2021-11-13
> **response to Reviewer KMX2**
>
>
> >> Concern about the new models
>
> We appreciate the suggestion of adapting more fancy structures. The bidirectional LSTM in our model is used to encode textual embeddings. And the Match module actually applies Attention Mechanism to process the information passed in. We train our model from scratch and achieve near perfect performance on the current task. Considering the trade off between the training expense and performance gain, we believe it's not necessary to apply more complicated structures for now.
>
> >> Concern about generalizing to other applications
>
> We focus on the reading-to-act task, which involves high-level instructions. In such a setting, the backwards manner is actually the natural and logical process of reasoning. To be specific, to fulfill the instruction, it needs to figure out the target monster; to beat the target monster, it needs to figure out the target weapon. Each reasoning result is the condition for the next reasoning process. When applying to other instruction following tasks, the logical process is in the same direction of the reasoning process. Our model will naturally output the sub-goals in a forward manner. The key insight of our model is following the logical process similar to human reasoning, which would be the backward manner in highly abstract tasks.
>
> >> Clarity
>
> We are sorry that the writing makes any confusion. To help readers fully understand the task, we tried to spend quite some space to describe the RTFM task. We will refine our paper in the revision and include more details of RTFM in the appendix, and also guide the readers to corresponding parts in the original RTFM paper (ICLR 2020).

---

> > ### Comment · Reviewer_KMX2 · 2021-11-20
> > **response**
> >
> > 1. The first part of the response is fair. Using the latest transformer models would have been the natural thing to do, and without further justification using a ad-hoc seeming matching model seems unnecessary but since it does do the job from the language side, I guess it's ok.
> >
> > For #2, I gave detailed reasons why I am worried that these results and experimental analyses are too specific to the RTFM task. I'm not sure that a concrete counter-reasoning was given. In particular this would require experimental verification:
> >
> > > Our model will naturally output the sub-goals in a forward manner
> >
> >  Though I agree there is some intuitive reason to believe it's likely.
> >
> > For #3, if a new revision is available I'd be happy to look and see if it reads better, but so far there doesn't seem to be one?

---

> > > ### Author Response · Authors · 2021-11-22
> > > **Thank you for your response**
> > >
> > > 1. We have submitted a new version of our draft. We really appericiate it if you can read it.
> > > 2. We are adding experiments to show the reasoning ability of our manager model. We will post the results later.

---

### Decision · Program_Chairs · 2022-01-20

**Decision:**

Reject

**Comment:**

The reviewers all raise critical issues with regard to both description and equations, and indicate that figures are not helpful. This is even after the revision. In response to KMX2, the authors suggested they will post additional experiments, but did not return to indicate that. This seems critical to answer empirical concerns about generality of the approach. We recommend to address these issues, if the authors decide to resubmit.

The meta review and recommendation discount the review of RC7c. Unfortunately, the AC and other reviewers, weren't able to engage RC7c in the discussion and the review was extremely short.